# De Novo Metastatic Prostate Cancer: Are We Moving toward a Personalized Treatment?

**DOI:** 10.3390/cancers15204945

**Published:** 2023-10-11

**Authors:** Claudia Piombino, Marco Oltrecolli, Elena Tonni, Marta Pirola, Rossana Matranga, Cinza Baldessari, Stefania Pipitone, Massimo Dominici, Roberto Sabbatini, Maria Giuseppa Vitale

**Affiliations:** 1Division of Oncology, Department of Oncology and Hematology, University Hospital of Modena, 41124 Modena, Italy; 256171@studenti.unimore.it (C.P.); 297143@studenti.unimore.it (M.O.); 325770@studenti.unimore.it (E.T.); 310546@studenti.unimore.it (M.P.); 297415@studenti.unimore.it (R.M.); baldessari.cinzia@aou.mo.it (C.B.); pipitone.stefania@aou.mo.it (S.P.); massimo.dominici@unimore.it (M.D.); sabbrob@unimore.it (R.S.); 2Laboratory of Cellular Therapy, Division of Oncology, Department of Medical and Surgical Sciences for Children and Adults, University of Modena and Reggio Emilia, 41124 Modena, Italy

**Keywords:** metastatic hormone-sensitive prostate cancer, new hormonal agents, transcriptomic profiling, DNA damage repair genes, tumor suppressor genes, androgen receptor, immunotherapy, CDK4/6 inhibitors, PARP inhibitors, AKT inhibitors

## Abstract

**Simple Summary:**

De novo metastatic hormone-sensitive prostate cancer usually has a dismal prognosis, which has slightly improved in recent years thanks to the introduction of new hormonal agents and chemotherapy combined with androgen deprivation therapy from the first-line setting. The randomized clinical trials that have furnished the current therapeutic options stratified patients according to clinical criteria that do not necessarily reflect the biological rationale of the chosen therapy. With the accumulation of data on genomic features and transcriptomic profiling, several ongoing clinical trials are investigating new therapeutic approaches and the efficacy of a biomarker-guided treatment with the aim of defining a personalized treatment for de novo metastatic hormone-sensitive prostate cancer.

**Abstract:**

De novo metastatic hormone-sensitive PC (mHSPC) accounts for 5–10% of all prostate cancer (PC) diagnoses but it is responsible for nearly 50% of PC-related deaths. Since 2015, the prognosis of mHSPC has slightly improved thanks to the introduction of new hormonal agents and chemotherapy combined with androgen deprivation therapy from the first-line setting. This review describes the current therapeutic opportunities for de novo mHSPC, focusing on potential molecular biomarkers identified in the main clinical trials that have modified the standard of care, the genomic features of de novo mHSPC, and the principal ongoing trials that are investigating new therapeutic approaches and the efficacy of a biomarker-guided treatment in this setting. The road toward personalized treatment for de novo mHSPC is still long, considering that the randomized clinical trials, which have furnished the basis of the current therapeutic options, stratified patients according to clinical criteria that did not necessarily reflect the biological rationale of the chosen therapy. The role of transcriptomic profiling of mHSPC as a predictive biomarker requires further validation, and it remains to be ascertained how the genomic variants detected in mHSPC, which are regarded as predictive in the castration-resistant disease, can be exploited in the mHSPC setting.

## 1. Introduction

According to GLOBOCAN 2020, almost one and a half million new cases of prostate cancer (PC) and approximately 400.000 PC-related deaths were reported in 2020 globally [1]. De novo metastatic hormone-sensitive PC (mHSPC) accounts for 5–10% of all PC diagnoses, but it is responsible for nearly 50% of PC-related deaths [2,3]. The incidence of de novo mHSPC is increasing in Western countries, probably due to the introduction of new diagnostic tools in the imaging of PC, such as PSMA-PET, and a reduction in PSA opportunistic screening [4,5,6]. De novo mHSPC is characterized by an aggressive course with a briefer time of onset of castration resistance and worse overall survival (OS) in contrast with metachronous mHSPC [7]. Since 2015, the prognosis of mHSPC has slightly improved thanks to the introduction of new hormonal agents (NHAs) and chemotherapy, combined with androgen deprivation therapy (ADT) from the first-line setting [8,9,10,11,12,13,14]. Nonetheless, the current therapeutic decision making for mHSPC, unlike in metastatic castration-resistant PC (mCRPC), is still based on clinical features (e.g., high-volume vs. low-volume disease, visceral vs. bone-only metastasis) since clinical trials evaluating molecular-biomarker-guided treatment of mHSPC are still ongoing.

This narrative review aims to describe the current therapeutic opportunities for de novo mHSPC, focusing on potential molecular biomarkers identified in the main clinical trials that have modified the standard of care (SOC), the genomic features of de novo mHSPC, and the main ongoing trials that are investigating new therapeutic approaches and the efficacy of a biomarker-guided treatment in this setting. The literature research was conducted by starting from the main clinical trials that evaluated chemotherapy and NHA for mHSPC, and then for each of them, searching for possible further analysis focused on de novo mHSPC and associated molecular biomarkers. A review of the principal works investigating the genomic features of de novo mHSPC compared with metachronous mHSPC and mCRPC was performed. Finally, research from https://clinicaltrials.gov (accessed on 1 September 2023) was conducted to identify ongoing phase III clinical trials testing new therapeutic approaches for mHSPC.

## 2. Current Therapeutic Opportunities for De Novo mHSPC

### 2.1. Doublet Therapy

#### 2.1.1. Docetaxel Plus ADT

The first study that redefined the treatment paradigm for mHSPC was the CHAARTED (ChemoHormonal Therapy versus Androgen Ablation Randomized Trial for Extensive Disease in Prostate Cancer) trial [8]. This randomized phase III trial study enrolled 790 patients affected by mHSPC (575 of them with de novo disease), with the aim to verify the superiority of upfront docetaxel 75 mg/mq given every 21 days for up to six cycles in association with ADT over ADT alone. After a median follow-up of 53.7 months, an absolute benefit in terms of the mOS of 16.8 months was observed in the experimental arm compared with ADT alone (mOS: 51.2 vs. 34.4 months, HR: 0.63, 95%CI: 0.50–0.79, *p* < 0.001) in patients with high-volume disease (as determined using the presence of at least four bone metastatic lesions with at least one beyond the vertebral bodies and pelvis and/or using evidence of visceral metastases), while no benefit was reported in men with low-volume mHSPC [15]. Transcriptional profiling of primary PC samples belonging to 160 men enrolled in this trial (of which 88% with synchronous mHSPC and 78% with high-volume disease) was performed by Hamid et al. [16] using the PAM50 classifier (luminal A, luminal B, and basal subtypes), the Decipher genomic classifier, and androgen receptor activity (AR-A, defined as average or lower) [17,18,19]. The analysis revealed a predominance of luminal B (50%) and basal (48%) subgroups, lower AR-A, and high Decipher risk tumors. The luminal B subgroup benefited significantly from the addition of docetaxel to ADT in terms of OS, while the basal subtype showed no OS advantage, even in the case of high-volume disease. In the multivariate analysis, higher Decipher risk and lower AR-A significantly correlated with poorer OS. Furthermore, the combination therapy conferred greater improvements in the OS in the presence of a higher Decipher risk. This study proposed both prognostic and predictive value of transcriptional subtyping for mHSPC.

#### 2.1.2. Abiraterone Plus ADT

The double-blind phase III trial LATITUDE [10] was the first study to demonstrate the benefit of an upfront combination therapy with an NHA. A total of 1199 patients affected by de novo high-risk mHSPC, which was defined by at least two out of three risk factors (Gleason score ≥ 8, at least three bone metastatic lesions, and the evidence of visceral metastasis), were 1:1 randomized to be treated with abiraterone acetate plus prednisone (or prednisolone) plus ADT versus placebo plus ADT. Considering the notable advantage in terms of the radiological progression-free survival (rPFS) and OS observed in the experimental arm at an interim analysis, the trial was subsequently unblinded and crossover was allowed. At the final OS analysis (median follow-up 51.8 months), 72 patients had crossed over to abiraterone acetate from the control group; the mOS was 53.3 months (95%CI: 48.2 months to not reached (NR)) in the experimental arm vs. 36.5 months (95%CI: 33.5–40.0 months) in the control group (HR: 0.66, *p* < 0.0001) [20]. No analysis of predictive biomarkers of response to abiraterone acetate was reported. An interesting multivariable model using data from the LATITUDE trial identified 11 prognostic variables commonly assessed in clinical practice (performance status, number of bone metastatic lesions, Gleason score, evidence of liver metastasis, worst pain score, albumin, LDH level, PSA level, hemoglobin level, and treatment regimen) that accurately predict prognosis and ameliorate risk stratification for de novo mHSPC [21].

#### 2.1.3. Enzalutamide Plus ADT

The role of the NHA enzalutamide associated with ADT as an upfront therapy for mHSPC was investigated in two phase III clinical trials. In the double-blind ARCHES trial [13], a total of 1150 patients with mHSPC were 1:1 randomized to receive enzalutamide plus ADT or placebo plus ADT. Previous treatment with docetaxel was allowed. Enzalutamide significantly decreased the risk of radiographic disease progression or death by 61% compared with ADT alone (HR: 0.39, 95%CI: 0.30–0.50, *p* < 0.001), irrespective of previous local and/or systemic treatment, disease volume, and risk [22]. A post hoc analysis demonstrated the clinical advantage of enzalutamide in both cases of de novo mHSPC and metachronous mHSPC [23]. After unblinding, 180 progression-free men assigned to the control arm crossed over to enzalutamide plus ADT. The final prespecified analysis of the OS (median follow-up 44.6 months) showed that enzalutamide decreased the risk of death by 34% compared with ADT alone (median NR in either group, HR: 0.66, 95%CI: 0.53–0.81, *p* < 0.001) [24].

In the ENZAMET study [12], a total of 1125 patients affected by mHSPC were 1:1 randomly assigned to be treated with enzalutamide or a non-steroidal first-generation anti-androgen (bicalutamide, flutamide, or nilutamide) in association with ADT. A total of 52% of the patients had high-volume disease. A total of 65% of patients in the enzalutamide group and 76% of patients in the control group received a prior course of six cycles of docetaxel. Furthermore, concurrent upfront docetaxel was permitted after a protocol amendment early during accrual. At the planned primary OS analysis (median follow-up 68 months), the mOS was NR in both groups (HR: 0.70, 95%CI: 0.58–0.84, *p* < 0.0001), with a 5-year OS of 57% in the control group and 67% in the enzalutamide group. The enzalutamide advantage in terms of the OS was consistent across the predefined prognostic subgroups (de novo vs. metachronous mHSPC, high-volume vs. low-volume disease) and in those who received concomitant docetaxel [25]. Unfortunately, no analysis of the predictive biomarkers of the response to enzalutamide was described.

#### 2.1.4. Apalutamide Plus ADT

The efficacy of the NHA apalutamide plus ADT compared with ADT plus placebo was assessed in the double-blind phase III trial TITAN [14]. Eligible patients had mHSPC with at least one lesion detectable on bone scanning; previous docetaxel therapy was allowed. Among the 1052 enrolled patients, 10.7% had received prior docetaxel chemotherapy and 62.7% had high-volume disease; more than 80% of patients had metastatic synchronous disease. A total of 40% of the patients in the control group crossed over to the experimental arm after the initial unblinding at 22.7 months of follow-up. At a median follow-up of 44 months, apalutamide in combination with ADT significantly decreased the risk of death by 35% compared with ADT alone (mOS: NR vs. 52.2 months, HR: 0.65, 95%CI: 0.53–0.79, *p* < 0.0001) and by 48% after adjusting for crossover. The subgroup analysis pointed out that a benefit from apalutamide was detected in almost all subgroups, notably in both cases of low- and high-volume mHSPC; a trend toward favoring the placebo in men who had received previous chemotherapy was registered, although these patients represented only 10% of the trial population and no interaction between the efficacy of apalutamide and prior docetaxel was detected in a post hoc interaction test [26]. In a post hoc analysis, similar to what was performed by Hamid et al. [16], the transcriptional profiling of primary PC samples from 222 patients enrolled in TITAN revealed that most patients had a high-Decipher-risk disease. In the control group, patients with a high Decipher risk had a poorer prognosis than those with a low-to-average Decipher risk, while no prognostic difference between these two different classes of risk was observed in the apalutamide group. Both basal and AR-A low subtypes showed significant benefit from apalutamide, suggesting that apalutamide is beneficial, especially for the highest-risk molecular subtypes [27]. A more persistent benefit of the addition of apalutamide to ADT in men with a high Decipher genomic classifier score compared with patients with a low genomic classifier score was also confirmed in a cohort of 233 patients from the SPARTAN trial [28]. In an exploratory analysis that investigated correlations between the biomarkers and OS in TITAN, the detection of circulating tumoral DNA (ctDNA) or any androgen receptor (AR) genomic alterations at baseline and any AR genomic alterations or PI3K pathway activation at the end of the study treatment were significantly associated with a poor OS in multivariate analyses from both treatment groups [29].

### 2.2. Triplet Therapy

More recently, the need for further treatment intensification with triplet therapy, consisting of the association of ADT with both docetaxel and NHA, was investigated by the phase III trials ARASENS and PEACE-1. ARASENS [30] enrolled 1306 patients affected by mHSPC that were eligible for ADT and chemotherapy with docetaxel to be treated with either darolutamide or a placebo in addition to docetaxel for six cycles and ADT. Most patients (86.1%) had de novo mHSPC. The primary analysis showed a 32.5% (HR: 0.68, 95%CI: 0.57–0.80, *p* < 0.001) lower risk of death in the darolutamide group than in the placebo one: with a median follow-up of 43.7 months in the experimental arm and 42.4 months in the placebo arm, the mOS was NR in the experimental group vs. 48.9 months in the control group. According to safety analyses, adverse events (AEs) of any grade were similar in both arms: the most common grade 3 or 4 AE was neutropenia associated with docetaxel. Post hoc analyses showed a significant OS benefit in favor of the addition of darolutamide in all patients, with more consistent outcomes in the high-volume (mOS: NR vs. 42.4 months, HR: 0.69, 95%CI: 0.57–0.82), high-risk (mOS: NR vs. 43.2 months, HR: 0.71, 95%CI: 0.58–0.86), and low-risk (mOS: NR vs. NR, HR: 0.62, 95%CI: 0.42–0.90) disease subgroups [31]. However, most of the patients included in the ARASENS trial had high-volume (77%) and/or high-risk (70%) mHSPC: the low-volume population was not well represented (only 23%). Thus, it is not possible to draw definitive conclusions for patients with low-volume mHSPC.

PEACE-1 [32] was a 2 × 2 factorial design trial that enrolled 1173 patients with de novo mHSPC. Eligible participants were therefore randomly assigned in a 1:1:1:1 manner to receive the SOC (ADT alone or with docetaxel for six cycles; the 2017 amendment made the association of both mandatory), SOC plus external beam radiotherapy (EBRT) to the primary tumor, SOC plus abiraterone in association with prednisone, or SOC plus abiraterone and EBRT to the primary tumor. To evaluate the efficacy of abiraterone in addition to SOC, on the basis of the assumption of the absence of significant interactions between abiraterone and EBRT to the primary tumor, they conducted a 2 × 2 factorial analysis. They pooled the groups 2 × 2, distinguishing those who received abiraterone with or without EBRT to the primary tumor into one group and comparing them to those who did not receive it (SOC with or without EBRT to the primary tumor). At a median follow-up of 3.5 years, the addition of abiraterone significantly increased the median rPFS (4.46 vs. 2.22 years, HR: 0.54, 95%CI: 0.41–0.71), with a reduction in the relative risk of radiographic progression by 46%. With a median follow-up of 4.4 years, a significant benefit in terms of the mOS was also reported for patients receiving abiraterone (5.72 vs. 4.72 years, HR: 0.82, 95%CI: 0.69–0.98, *p* = 0.03), with a risk of death from any cause being 18% lower than in those who did not receive it. The effect of abiraterone was particularly marked in men with high-volume mHSPC (median rPFS: 4.46 vs. 2.03 years, HR: 0.50; mOS: NR vs. 4.43 years, HR: 0.75). From the safety point of view, abiraterone did not produce a significant increase in neutropenia, febrile neutropenia, fatigue, or neuropathy rates compared with ADT plus docetaxel alone; the only exceptions were hypertension, hypokalemia, and higher levels of aminotransferases, which were more frequently reported in the group treated with abiraterone.

### 2.3. How to Currently Choose the Most Suitable Treatment for Each Patient

Both ARASENS and PEACE-1 showed that upfront treatment intensification with the combination of ADT, docetaxel, and NHA for de novo mHSPC could become the new SOC since it improved survival outcomes with an acceptable safety profile, especially in patients with high-volume symptomatic disease and without severe comorbidities and a long life expectancy. Probably, the association of the NHA acts as a maintenance treatment, prolonging the effect of chemotherapy. However, no predictive biomarker of response to triplet therapy has been reported up to now.

In patients without a high metastatic burden nor symptomatic disease, considering the lack of robust predictive biomarkers and the different inclusion criteria adopted in each pivotal trial, the choice of the most suitable treatment is currently based mainly on clinical aspects. Abiraterone is known to have pronounced cardiovascular side effects due to the associated increase in mineralocorticoid production [33]. On the other hand, both enzalutamide and apalutamide are potent inducers of CYP3A4 and may increase the risk of mental impairment, and thus, they are potentially harmful to elderly people suffering from multiple pathologies, functional or cognitive impairment, and/or behavioral alterations [34]. Even if the current tendency to intensify treatment has a solid scientific rationale derived from the aforementioned clinical trials, PC remains mainly a disease of the elderly. Consequently, treatment has to be tailored by considering the patient’s life expectancy, comorbidities, and the presence of symptoms while trying to preserve the quality of life.

### 2.4. Oligometastatic Prostate Cancer

Oligometastatic PC (omPC) encompasses a heterogeneous group of tumors characterized by a low metastatic burden [35]. While some works defined omPC based on the number of metastatic lesions, ranging from 3 to 5 lesions, other authors adopted the criteria of low-volume disease according to the CHAARTED trial [8] or low-risk disease according to the LATITUDE trial [10] for the definition of omPC as either de novo or recurrent [36]. Considering that de novo omPC generally displays indolent behavior, with node metastases only or limited bone involvement, and it is associated with a better prognosis compared with men with more than five lesions [37], a benefit from different treatment options may be observed. In fact, post hoc analysis of the CHAARTED [8] and GETUG-AFU15 trials [38] showed that patients with low-volume disease had a much longer OS, without evidence that docetaxel improved OS, irrespective of whether patients received ADT plus docetaxel for de novo mHSPC or after prior local treatment [39]. Contrariwise, a post hoc analysis of the STAMPEDE trial arm G [11] demonstrated that adding abiraterone to ADT also improves the OS in low-volume de novo mHSPC (HR: 0.60, 95%CI: 0.39–0.92) [40]. Similarly, upfront enzalutamide or apalutamide conferred a disease-burden-independent advantage over ADT alone in phase III pivotal studies [12,13,14]. 

Different therapeutic approaches for de novo omPC include locoregional treatments, mainly radiation therapy. In the HORRAD trial [41], 432 patients with primary bone mHSPC were randomized to receive only ADT or ADT in combination with EBRT to the primary tumor; the subgroup analysis demonstrated a trend toward an OS benefit only in men with fewer than five skeletal lesions (HR: 0.68, 95%CI: 0.42–1.10). These promising results were further investigated in the STAMPEDE trial arm H [42]: EBRT to the prostate significantly improved the OS in patients with low metastatic load according to the CHAARTED criteria (HR: 0.68, 95%CI: 0.52–0.90, *p* = 0.007), reporting an increase in the 3-year survival rate from 73% to 81% with EBRT. In a recent phase II trial including 200 men with de novo omPC (as determined using the presence of five or fewer skeletal or extrapelvic nodal metastases and the absence of visceral metastases) randomized to receive either ADT or ADT plus radical local treatment on the prostate, both the rPFS and OS were significantly improved in the experimental arm [43]. However, the opposite results were recently presented at the last ASCO genitourinary symposium from the PEACE-1 trial [30]: in men with de novo low-volume mHSPC (at most three bone metastases with or without nodal involvement), combining prostate EBRT to systemic treatment did not improve the OS [44]. The differences that emerged in these trials were probably due to the different definitions of low-volume diseases, as well as the different systemic treatments administered to the patients. Nevertheless, EBRT to the prostate in association with systemic treatment is recommended for men with low-volume mHSPC according to the ESMO and NCCN guidelines [45,46].

In addition to EBRT to the primary tumor, metastasis-directed therapy (MDT) is a debated issue. MDT is generally used to treat bone metastases or pathological lymph nodes. The only two prospective trials that investigated stereotactic ablative radiotherapy (SABRT) versus observation, namely, STOMP and ORIOLE, were focused only on metachronous omPC and demonstrated that MDT prolongs the androgen-deprivation-free survival and PFS more than observation alone [47,48]. Although for de novo omPC, there is no randomized trial evidencing a benefit from MDT of all documented lesions, there is a strong consensus for a combined approach (ADT plus additional systemic therapy, local radiotherapy, and MDT) [49]. Current evidence was derived from various case series that investigated a combined approach with encouraging results [50,51,52,53]. Many trials are ongoing to define whether the combination of ADT plus SABRT for de novo omPC improves outcomes compared with systemic treatment alone (NCT03298087, NCT05707468, NCT04983095, NCT04115007, NCT05223803, NCT04619069, NCT03784755, NCT05212857, NCT05209243).

Adding radiation therapy to systemic treatment has a potential biological rationale: radiotherapy induces cell death, and the dying cells release “danger signals” that, in turn, might make cancer cells outside the radiation field more susceptible to an immune-mediated cytotoxic environment (the so-called abscopal effect) [54]. Moreover, radiation therapy might prevent metastasis-to-metastasis spread. Characterizing multiple metastases arising from PC in ten patients under ADT with whole-genome sequencing, Goundem et al. [55] demonstrated the existence of metastasis-to-metastasis spread, either via de novo monoclonal seeding of daughter metastases or via the movement of multiple tumor clones between metastatic sites.

Although MDT seems to be effective for omPC, little is known about the predictive biomarkers of response to the different treatment options available in this setting [56,57]. The study of predictive biomarkers might be useful for identifying which patients could benefit from ADT only or ADT combined with chemotherapy, NHA, and/or local treatments. The only data available were derived from a pooled analysis of STOMP and ORIOLE trials, where the largest benefit of MDT for metachronous omPC was observed in patients with high-risk pathogenic somatic variants within *ATM*, *BRCA1*/*2*, *Rb1*, or *TP53*, suggesting that a high-risk mutational signature may differentiate the treatment response after MDT [58].

## 3. Genomic Features of mHSPC

The aim of therapy modulation and personalization for de novo mHSPC may be reached via the study of biology and biomarkers. However, the mutation profile of mHSPC is poorly characterized since sequencing efforts have focused on either localized PC or mCRPC. Progression from localized PC to mCRPC is characterized by the accumulation of deleterious genomic mutations in the latter disease state. In detail, the most frequently altered genes in mCRPC are tumor suppressor genes (*RB1*, *TP53*, and *PTEN*) and genes involved in the androgen receptor (AR) pathway, chromatin remodeling (*KMT2C* and *KMT2D*), PI3K signaling (*AKT1* and *PIK3CA*), and DNA damage repair (DDR) (*BRCA2*, *BRCA1*, *ATM*, and *FANCA*) [59,60,61]. Different data seem to indicate that the mutation profile of mHSPC lies between localized PC and mCRPC, suggesting that the enrichment of deleterious alterations over time confers a survival advantage to cancer cells, inducing treatment resistance [62,63]. A systematic metanalysis [56] that included 1682 mHSPC patients, of whom 1248 (74%) had de novo disease, from 11 studies pointed out that the most commonly mutated genes in terms of mutations or copy number alterations were *TP53* (32%) and *PTEN* (20%), followed by genes involved in DDR (18%), with *BRCA2* as the most frequently mutated gene (7%); alterations in cell cycle signaling were reported in 7–13% of the cases. Tumors from men with de novo mHSPC were enriched with mutations in *TP53* and *CDK12* compared with metachronous mHSPC, while cell cycle signaling, Wnt pathway, *PTEN*, and *SPOP* alterations were more frequent in metachronous mHSPC. In high-volume disease according to the CHAARTED criteria [8], *TP53*, *BRCA2*, *PIK3CA*, *RB1*, and *APC* were more frequently altered compared with low-volume disease. However, the DNA source and definitions for gene alterations differed significantly between studies, including somatic alterations from formalin-fixed paraffin-embedded (FFPE) material, as well as ctDNA. Among the studies included in the aforementioned metanalysis, a noteworthy observation was derived from the targeted next-generation sequencing (tNGS) performed on 185 tumor samples obtained almost entirely from de novo mHSPC patients enrolled in the STAMPEDE trial: PI3K pathway aberrations were observed in 43% of the cases, which were due to *PTEN* copy-number loss (34%) and/or inactivating mutations in *PIK3* or *AKT* (18%) [64].

### 3.1. The Role of Liquid Biopsy

Most men with de novo mHSPC will not receive primary surgery; the histologic diagnosis is typically performed on a prostatic biopsy. Therefore, a liquid biopsy could add clinically relevant information in this setting. In a single-center prospective cohort, Vanderkerhove et al. [65] detected a median plasma ctDNA fraction of 11% (range 2.0–84%) among 26 out of 35 (74%) untreated patients with de novo mHSPC; for the remaining 9 patients, ctDNA was not detectable. Higher ctDNA levels were identified in the presence of visceral metastasis. A somatic analysis of ctDNA and tumor tissue revealed a mutational landscape like mCRPC, although without *AR* gene alterations: *TP53* and DDR gene mutations were identified in 47% and 21% of the cases, respectively. The rate of concordance for mutation detection between tumor tissue and ctDNA was 80%, suggesting that de novo mHSPC was a highly clonal disease at diagnosis. On the other hand, in a cohort of 82 Chinese patients with de novo mHSPC, only 50% of men had a ctDNA fraction >2% and the percentage of ctDNA-positive patients was even lower (37%) in a cohort of 73 untreated mHSPC, including both de novo and metachronous disease [66,67]. There are still some issues to solve prior to introducing liquid biopsy technology in routine clinical practice regarding preanalytical aspects and low-circulating tumor content, considering that common PC copy number variants, such as *PTEN* or *CDH1* deletions, are undetectable in the presence of a low ctDNA fraction [68]. Consequently, in men with a low ctDNA fraction, tissue biopsy profiling remains more informative.

### 3.2. Prognostic Information

From a prognostic point of view, data regarding the association between the genetic alterations, time to castration resistance, and OS for de novo mHSPC are partial because they were obtained from cohorts including both synchronous and metachronous metastatic disease. Among 424 cases of mHSPC, including 275 men with de novo mHSPC, Stopsack et al. [69] reported a rate of progression to castration resistance that was 1.6-to-5-fold higher in the presence of alterations in *AR*, *TP53*, cell cycle, and MYC pathways and approximately 1.5-fold lower with *SPOP* and Wnt pathway alterations; similarly, the OS rate was 2-to-4-fold higher in the presence of *AR* or cell cycle alterations, and 2-to-3-fold lower if the *SPOP* or Wnt pathway was altered. The sequencing of FFPE tissue from biopsies of 43 patients affected by mHSPC, of whom 30 had de novo disease, revealed a slightly poorer OS with cumulative mutations or alterations in the tumor suppressor genes *TP53*, *PTEN*, and *RB1* [63]; the negative prognostic value of alterations in *TP53*, *PTEN*, and *RB1* was also observed in a cohort of 97 men with mHSPC treated with first-line ADT plus docetaxel or abiraterone acetate, outperforming clinical criteria that predict early disease progression [70]. An association between a shorter OS and alterations in *TP53*, *ATM*, and DDR genes detected on plasma ctDNA was also observed among 53 patients with de novo or metachronous untreated mHSPC [67]. Finally, tNGS across 113 genes performed on 202 primary tumor samples obtained from patients with synchronous or metachronous mHSPC revealed a significantly shorter OS in the presence of mutations or deep deletions of *RB1* [71].

The association between *SPOP* mutations and better prognosis was also detected in a cohort of 121 patients with de novo mHSPC treated with ADT: both the median PFS and OS were significantly improved in the subset of 25 men with *SPOP*-mutated cancers (mPFS: 35 vs. 13 months, *p* = 0.016; mOS: 97 vs. 69 months, *p* = 0.027) [72]. The SPOP protein is involved in the ubiquitination and consequent proteasomal degradation of target proteins; in PC, SPOP acts as a tumor suppressor by targeting several proteins, including AR, SRC3, and BRD4 [73]. The hypothesis that *SPOP*-mutated PC is primarily driven by AR signaling was tested in a real-world setting: in a cohort of men with de novo mHSPC undergoing ADT plus NHA, the presence of *SPOP* mutation compared with wild-type was associated with a longer time to castration resistance and OS, while the *SPOP* mutational status was not associated with the time to castration resistance or OS in a cohort treated with ADT plus docetaxel [74]. *SPOP* mutation may therefore be used as a predictive biomarker to guide the treatment choice for patients with de novo mHSPC.

## 4. Ongoing Phase III Clinical Trials Testing New Therapeutic Approaches for mHSPC

Apart from trials focusing on NHA, ADT, and chemotherapy with different schedules for mHSPC (ARANOTE NCT04736199, ARASAFE NCT05676203, LIBERTAS NCT05884398, NCT05956639), several other ongoing phase III clinical trials are investigating the role of new therapeutic approaches (immunotherapy, radiopharmaceuticals, and molecular target agents) in this setting (Table 1, Figure 1).

### 4.1. Immunotherapy

Although the expression of programmed death ligands 1 and 2 (PD-L1 and PD-L2) on PC cells is highly variable, therapy with enzalutamide can upregulate PD-L1 expression in the tumor microenvironment; this can represent a mechanism of resistance by inducing immune evasion [75]. In the phase Ib Keynote-028 and phase II Keynote-199 trials, mCRPC enzalutamide-refractory patients and previously untreated patients received a combination of pembrolizumab and enzalutamide, reaching potentially improved and durable response rates [76,77]. Based on these premises, the ongoing randomized, double-blind, placebo-controlled phase III KEYNOTE-991 (NCT04191096) [78] is investigating whether this combination therapy for NHA-naive patients with mHSPC is superior to enzalutamide plus placebo. Stratification by prior docetaxel therapy and the presence of high-volume mHSPC is planned. Pembrolizumab 200 mg every three weeks will be administered for up to 35 cycles, loss of clinical benefit, or intolerable AEs. The two co-primary endpoints are the OS and rPFS. Archival or newly obtained tumor tissue and blood for genetic, RNA, serum, and plasma biomarkers and ctDNA analyses will be collected from all participants to support exploratory analyses of novel biomarkers. PROSTRATEGY (NCT03879122) is another phase III clinical trial that is investigating the role of immunotherapy for high-volume mHSPC [79]. This trial will randomize approximately 135 patients into three arms: ADT + docetaxel for six cycles (control arm, ARM 1); ADT + docetaxel for six cycles and then nivolumab 3 mg/kg every 14 days for one year (ARM 2); and ADT + two cycles of ipilimumab 3 mg/kg every 3 weeks, followed by three cycles of docetaxel, two cycles of ipilimumab, three cycles of docetaxel, and then nivolumab 3 mg/kg every two weeks for 12 months (ARM 3). The primary endpoint will be the OS.

### 4.2. Radiopharmaceuticals

Lutetium-177(177Lu)-PSMA-617 is a beta emitter radioisotopic agent that was approved by the FDA in 2022 for the treatment of mCRPC in men who had progressed to an NHA and taxane-based chemotherapy, and whose metastatic lesions express the prostatic-specific membrane antigen (PSMA), as documented via PSMA imaging [80]. Radiopharmaceuticals release alpha or beta radiation to cancer cells via radioisotopes; radiation activates apoptosis via single- and double-strand DNA breaks [81]. PSMAddition (NCT04720157) [82] is a phase III, randomized, open-label, international, prospective clinical trial that aims to evaluate the efficacy and safety of 177Lu-PSMA-617 in combination with the SOC (ADT plus NHA) versus SOC alone for mHSPC. About 1126 patients will be randomized 1:1 to receive the SOC, with or without 177Lu-PSMA-617 administered once every 6 weeks for six cycles. The exclusion criterion will be a rapidly progressing tumor that requires chemotherapy. The primary endpoint will be the rPFS. Stratification according to age (≥70 years/<70 years), high-volume vs. low-volume disease, and prior/planned prostatectomy or radiotherapy of the prostate is planned.

### 4.3. Molecular Target Agents

The role of molecular target agents has been largely investigated in the mCRPC setting in combination with ADT. The increase in knowledge of the mutational profile in mHSPC is also leading to test-targeted treatments, such as cyclin-dependent kinase 4 and 6 (CDK4/6) inhibitors, poly(ADP-ribose) polymerase inhibitors (PARPi), and AKT-inhibitors (AKTi), in this setting [83].

#### 4.3.1. CDK4/6 Inhibitors

During the G1-S checkpoint, CDK4/6 activation by the AR axis contributes to cancer cell proliferation; among the mechanisms of resistance to NHA, the upregulation of cyclin D1 (whose association with CDK4/6 is crucial for the transition from G1 to S phase) was described [84]. CYCLONE-03 (NCT05288166) [85] is a placebo-controlled phase III study that will randomize about 900 patients affected by high-risk NHA-naïve mHSPC (defined by at least four bone metastasis and/or visceral disease) to receive either abemaciclib (a selective CDK4/6 inhibitor) or a placebo, plus abiraterone and prednisone. Visceral metastases and de novo mHSPC will be the stratification factors. The primary endpoint will be the rPFS.

#### 4.3.2. PARP Inhibitors

Preclinical and clinical evidence showed that the co-inhibition of the AR axis and PARP generates a combined anti-tumor effect: PARP is involved in the positive co-regulation of AR signaling, and thus, PARP/AR signaling co-inhibition leads to enhanced AR target gene suppression; moreover, treatment with NHAs inhibits the transcription of some DDR genes, inducing synthetic lethality due to cancer cells’ inability to repair DNA, even in patients without any DDR alterations [86]. The combination of the PARPi olaparib with abiraterone is FDA- and EMA-approved as a first-line treatment of mCRPC if chemotherapy is not clinically indicated, according to the results of PROPEL [87]. The combination of the PARPis talazoparib and enzalutamide was recently FDA-approved as a first-line treatment for men with homologous recombination repair (HRR) gene-mutated mCRPC, according to TALAPRO-2 [88]. The association of the PARPi niraparib with abiraterone was evaluated both in patients with and without HRR gene-altered mCRPC in the phase III MAGNITUDE trial [89]. Differently from PROPEL, the superiority of the experimental treatment over the control arm (abiraterone plus placebo) was demonstrated only in the cohort with HHR gene-altered mCRPC, while in the HRR proficient cohort, futility was confirmed per the prespecified criteria.

The HRR-related genes’ mutational status was determined with different technologies in these studies and the panel of genes tested varied among them. For PROPEL, both tumor tissue (FoundationOne CDX) and ctDNA-based (FoundationOne Liquid CDx) tests were employed to detect pathogenic variants in the assessed genes (*ATM*, *BRCA1*, *BRCA2*, *BARD1*, *BRIP1*, *CDK12*, *CHEK1*, *CHEK2*, *FANCL*, *PALB2*, *RAD51B*, *RAD51C*, *RAD51D*, and *RAD54L*) and a post hoc analysis combining tumor tissue and ctDNA data was carried out to increase the number of patients with a deficit in HRR and to at least reduce false negatives [87]. In TALAPRO-2, patients were prospectively evaluated for HHR-related gene (*ATM*, *ATR*, *BRCA1*, *BRCA2*, *CDK12*, *CHEK2*, *FANCA*, *MLH1*, *MRE11A*, *NBN*, *PALB2*, *RAD51C*) alterations in tumor tissue using FoundationOne CDx; a subsequent protocol amendment permitted prospective ctDNA testing using FoundationOne Liquid CDx [88]. Finally, in the MAGNITUDE trial, patients were prescreened for HRR status by testing both tissue and plasma with the FoundationOne CDx tissue test, Resolution Bioscience homologous recombination deficiency plasma test, or AmoyDx blood and tissue assays; to be considered HRR deficient, patients needed to have at least one gene alteration among *ATM*, *BRCA1*, *BRCA2*, *BRIP1*, *CDK12*, *CHEK2*, *FANCA*, *HDAC2*, and *PALB2* detected in at least one assay [89].

TALAPRO-3 (NCT04821622) [90] is a randomized double-blind trial that has recruited 599 men with mHSPC and HRR-related gene alterations (*ATM*, *ART*, *BRCA1*, *BRCA2*, *CDK12*, *CHEK2*, *FANCA*, *MLH1*, *MRE11A*, *NBN*, *PALB2*, *RAD51C*) to receive enzalutamide in association with placebo or talazoparib. The primary endpoint will be the rPFS. Patients will be stratified according to non-*BRCA* vs. *BRCA* alteration, low-volume vs. high-volume disease, and de novo vs. metachronous mHSPC. Similarly, the randomized, placebo-controlled, double-blind trial AMPLITUDE (NCT04497844) [91] has recruited 696 patients with mHSPC and HRR alterations to receive the PARPi niraparib or a placebo in combination with abiraterone. The primary endpoint will be the rPFS. Patients will be stratified according to disease volume, previous docetaxel-based chemotherapy, and the type of HRR-related gene defect.

#### 4.3.3. AKT Inhibitors

Inactivation of the tumor suppressor gene *PTEN* via deletion or mutation is frequent in PC, especially in late-stage tumors. PTEN loss of function determines PI3K/AKT signaling pathway activation and suppression of AR transcriptional output. AKTi activates AR signaling, suggesting the potential efficacy of the inhibition of both PI3K and AR signaling pathways [92]. Evidence supporting this association came from the phase III trial IPATential150 [93], which demonstrated that the AKTi ipatasertib, in association with abiraterone, improved the rPFS in patients with mCRPC and PTEN-loss. CAPItello-281 (NCT04493853) [94], which is a randomized double-blind trial, will test the AKTi capivasertib. Approximately 1000 patients with PTEN-deficient mHSPC, as demonstrated using tissue immunohistochemistry (IHC), will be randomized 1:1 to receive capivasertib or a placebo in association with abiraterone. The primary endpoint will be the rPFS.

However, assessing PTEN loss is a crucial issue to solve. In IPATential150, PTEN loss was assessed using IHC with a validated assay (VENTANA PTEN [SP218] assay) based on the evidence from a phase II study where the three methods employed (IHC, fluorescence in situ hybridization (FISH), and NGS) showed high concordance [95]. In detail, PTEN loss was determined via IHC using the absence of PTEN staining in 50% or more of the specimen’s tumor area; NGS to examine PTEN status or *PIK3CA*/*AKT1*/*PTEN* alteration was performed using a FoundationOne CDx NGS assay [93]. In a large radical prostatectomy cohort, 93% of tumors with intact PTEN according to IHC had no *PTEN* deletion in FISH, but the positive predictive value of IHC *PTEN* deletion was only 66% [96]. Different factors must be considered when PTEN loss is assessed: intratumoral heterogeneity; alterations that are undetectable by FISH, such as truncating mutations, structural rearrangements, epigenetic alterations, or post-transcriptional modifications; and intrinsic technical issues. IHC has the advantage of being rapid, inexpensive, and able to detect PTEN loss that is not caused by genetic alteration, particularly in the case of PTEN loss of heterozygosity [97].

## 5. Conclusions

The road toward personalized treatment for de novo mHSPC is still long, considering that the randomized clinical trials, which have furnished the basis of the current therapeutic options, stratified patients according to clinical criteria that did not necessarily reflect the biological rationale of the chosen therapy. Transcriptomic profiling of mHSPC revealed a predominance of aggressive and poor prognosis subtypes, but its role as a predictive biomarker requires further validation. Even though many of the genomic alterations detected in mHSPC are regarded as predictive in mCRPC, it remains to be ascertained how these alterations can be exploited in the mHSPC setting. In this sense, the ProBio (NCT03903835) trial, which is randomizing both mHSPC and mCRPC to receive SOC following national guidelines (control arm) or therapies based on a biomarker signature obtained from diagnostic tissue or liquid biopsy profiling (experimental arm), will probably provide a prospective evaluation of biomarker-driven treatments.

## Figures and Tables

**Figure 1 cancers-15-04945-f001:**
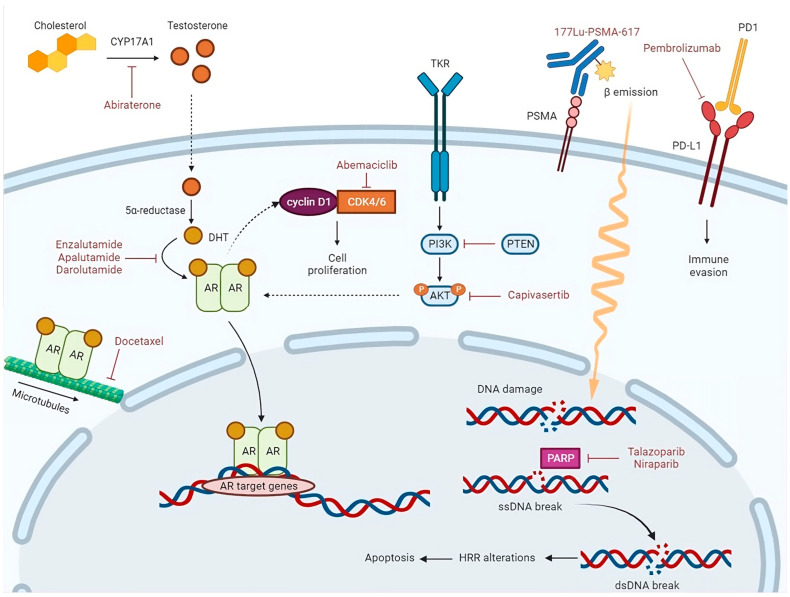
Abiraterone is a CYP17A1 inhibitor. CYP17A1 is a key enzyme in the steroidogenic pathway that produces testosterone. Testosterone is metabolized to dihydrotestosterone (DHT) by the enzyme 5α-reductase. The androgen receptor (AR), activated via binding of DHT in the cytoplasm, translocates into the nucleus, where it acts as a DNA-binding transcription factor that regulates AR target gene expression. Enzalutamide, apalutamide, and darolutamide competitively inhibit DHT binding to the AR, nuclear translocation of the AR, and DNA binding. Docetaxel inhibits AR nuclear translocation by targeting AR association with microtubules. During G1-S checkpoint, AR can bind to and activate cyclin D1, which by association with the cyclin-dependent kinases 4 and 6 (CDK4/6), contributes to cancer cell proliferation. Abemaciclib, which is a CDK4/6 inhibitor, arrests cell cycle and inhibits tumor growth. PI3K/AKT pathway is activated by binding of growth factors to tyrosine kinase receptor (TKR). Through phosphorylation, AKT controls activation or inactivation of various proteins involved in cell growth and proliferation. PTEN is the main downregulation protein of this pathway. AKT regulates transcriptional activity of the AR. Capivasertib, which is an AKT inhibitor, reduces AKT substrate phosphorylation and cell proliferation. The binding of the radioligand Lutetium-177(177Lu)-PSMA-617 to the prostate-specific membrane antigen (PSMA) results in its internalization and delivery of β-radiation into the cancer cells; radiation activates apoptosis via single-strand (ssDNA) and double-strand DNA (dsDNA) breaks. When an ssDNA break occurs, PARP recruitment and activation lead to DNA repair. In the presence of a PARP inhibitor, such as Talazoparib or Niraparib, unrepaired ssDNA breaks lead to dsDNA breaks during DNA replication. In cells with homologous recombination repair (HRR) alterations, dsDNA breaks are repaired by the error-prone non-homologous end-joining pathway, thus inducing genomic instability and consequent apoptosis.

**Table 1 cancers-15-04945-t001:** Ongoing phase III clinical trials testing new therapeutic approaches for mHSPC.

Official Title NCT Number	Control Arm	Experimental Arm(s)	Primary Endpoints	Status	Enrolment	Study Start/Completion Date
KEYNOTE-991 NCT04191096	Placebo + Enzalutamide + ADT	Pembrolizumab + Enzalutamide + ADT	OS, rPFS	Active, not recruiting	1251 (actual)	25 May 2021/ 2 February 2026
PROSTRATEGY NCT03879122	Arm 1: ADT + Docetaxel for 6 cycles	Arm 2: ADT + Docetaxel for 6 cycles and then Nivolumab 3 mg/kg every 14 days for one year Arm 3: ADT + 2 cycles of Ipilimumab 3 mg/kg every 21 days, followed by 3 cycles of Docetaxel, 2 cycles of Ipilimumab, 3 cycles of Docetaxel, and Nivolumab 3 mg/kg every 14 days for one year	OS	Active, not recruiting	135 (estimated)	11 February 2019/ 31 December 2024
PSMAddition NCT04720157	NHA + ADT	7.4 GBq (±10%) 177Lu-PSMA-617 once every 6 weeks (±1 week) for 6 cycles + NHA + ADT	rPFS	Recruiting	1126 (estimated)	9 June 2021/ 11 February 2026
CYCLONE-03 NCT05288166	Placebo + Abiraterone + Prednisone/Prednisolone	Abemaciclib + Abiraterone + Prednisone/Prednisolone	rPFS	Recruiting	900 (estimated)	14 April 2022/ 1 October 2027
TALAPRO-3 NCT04821622	Placebo + Enzalutamide	Talazoparib + Enzalutamide	rPFS	Active, not recruiting	599 (actual)	12 May 2021/ 10 April 2027
AMPLITUDE NCT04497844	Placebo + Abiraterone + Prednisone/Prednisolone	Niraparib + Abiraterone + Prednisone/Prednisolone	rPFS	Recruiting	696 (actual)	23 September 2020/ 27 May 2027
CAPItello-281 NCT04493853	Placebo + Abiraterone + Prednisone/Prednisolone	Capivasertib + Abiraterone + Prednisone/Prednisolone	rPFS	Recruiting	1000 (estimated)	13 July 2020/ 10 March 2026

OS: overall survival; rPFS: radiographic progression-free survival; ADT: androgen deprivation therapy; GBq: gigabecquerel; NHA: new hormonal agent.

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
