# Peer review of "De Novo Metastatic Prostate Cancer: Are We Moving toward a Personalized Treatment?"

_cancers, 2023, doi:10.3390/cancers15204945_

Round 1

Reviewer 1 Report

Metastatic castration-sensitive prostate cancer is commonly classified into clinical categories (high- vs low-volume disease, and de novo vs primary progressive disease) with a different prognosis and response to therapy. However, less is known about any underlying molecular features of mCSPC. This review is very complete as it aims to explore different molecular pathways and potential biomarkers to predict clinical outcome. In addition, the current work is very helpful providing new insights in precision medicine in mCSPC with ongoing clinical trials and molecular studies

The quality of English Language is good

Author Response

Many thanks for the feedback received.

Reviewer 2 Report

It has been a privilege to review this article. Overall, this is a very well-written, informative review article. This paper will help clarify the role of the new therapeutic approaches and the efficacy of a biomarker-guided treatment in “de novo” metastatic hormone-sensitive prostate cancer. I don't have any major recommendation or changes that need to be made.

1)However, it will improve the quality of the manuscript in point 4.3.2 to refer to the MAGNITUDE study and develop the different molecular analysis strategies that have been carried out in MAGNITUDE, TALAPRO-2 and PROPEL trials. 

2)Point out that the recruitment of the study AMPLITUDE has already been closed, it would be advisable to update.

3)Develop in point 4.3.3, the difficulties that exist when assessing the loss of PTEN between the different existing approaches, NGS and different IHC techniques in the published results of the IPATential150 trial and that may influence the results of ongoing studies with this approach.

Author Response

Many thanks for the feedback received.

  1. The MAGNITUDE study has been added to point 4.3.2 and the different molecular analysis performed in PROPEL, TALAPRO-2 and MAGNITUDE have been described.
  2. The data regarding the current recruitment status of AMPLITUDE have been updated.
  3. The available methods to assess PTEN loss have been illustrated in point 4.3.3

Reviewer 3 Report

Comments:

The review on de novo metastatic prostate cancer presents the current literature in the context of personalised therapy. It is a topical review which deserves attention. However, the review reads like a listing of studies / clinical trials highlighting their results and lacks insight from the authors. Below are my comments to improve the readability of the paper:

(1)   How was the literature research conducted? From its structure, this is not a systematic review of the literature; it reads more like a narrative review. Still, a few sentences about the methodological approach towards literature search must be mentioned after the Introduction.

(2)   A more critical approach towards each treatment technique is required to illustrate their advantages and drawbacks. For instance, after section 2 on Docetaxel+ADT; Abiraterone+ADT; Enzalutamide+ADT; Apalutamide+ADT (this should be subsection 2.1.4 not 2.1.3) the authors should add a concluding, comparative paragraph.

(3)   A paragraph on the limitations of current studies / trials is needed to identify gaps in research in order to submit recommendations for future trial designs.

(4)   Generally, more input from the authors is required to transform this review from a listing of studies into an analysis of the current treatment approaches for de novo metastatic prostate cancers.

Author Response

Many thanks for the feedback received.

  1. The methodological approach to select literature for this narrative review have been explained after the introduction
  2. A concluding paragraph entitled "How to choose now the most suitable treatment for each patient?" has been added.
  3. The limits of the available studies, in detail the absence of predictive biomarkers and the inclusion/exclusion criteria based only on clinical aspects is a stressed issue in all the review; adding a paragraph to further emphasize this limitation is redundant and does not reflect the aim of this review.
  4. The aim of this review is not to analyse the current treatment approaches for de novo metastatic prostate cancers, but to identify potential predictive biomarkers that can help to choose the most suitable treatment for each patient; the paragraph entitled "How to choose now the most suitable treatment for each patient?" provides author insights on the current available therapeutic options.

Round 2

Reviewer 3 Report

The authors have satisfactorily addressed the comments raised by this reviewer.